# ROBUSTNESS TO PRUNING PREDICTS GENERALIZATION IN DEEP NEURAL NETWORKS

## ABSTRACT

Why over-parameterized neural networks generalize as well as they do is a central concern of theoretical analysis in machine learning today. Following Occam's razor, it has long been suggested that *simpler* networks generalize better than more complex ones. Successfully quantifying this principle has proved difficult given that many measures of simplicity, such as parameter norms, grow with the size of the network and thus fail to capture the observation that larger networks tend to generalize better in practice. In this paper, we introduce a new, theoretically motivated measure of a network's simplicity: the smallest fraction of the network's parameters that can be kept while pruning without adversely affecting its *training loss*. We show that this measure is highly predictive of a model's generalization performance across a large set of convolutional networks trained on CIFAR-10. Lastly, we study the mutual information between the predictions of our new measure and strong existing measures based on models' margin, flatness of minima and optimization speed. We show that our new measure is similar to – but more predictive than – existing flatness-based measures.

## 1 INTRODUCTION

The gap between learning-theoretic generalization bounds for highly overparameterized neural networks and their empirical generalization performance remains a fundamental mystery to the field (Zhang et al., 2016; Jiang et al., 2020; Allen-Zhu et al., 2019). While these models are already being successfully used in many applications, improving our understanding of how neural networks perform on unseen data is crucial for safety-critical use cases. By understanding which factors drive generalization in neural networks we may further be able to develop more efficient and performant network architectures and training methods.

Numerous theoretically and empirically motivated attempts have been made to identify *generalization measures*, that is, properties of the trained model, training procedure and training data that distinguish models that generalize well from those that do not (Jiang et al., 2020). A number of generalization measures have attempted to quantify Occam's razor, i.e. the principle that *simpler* models generalize better than complex ones (Neyshabur et al., 2015; Bartlett et al., 2017). This has proven to be non-trivial, as many measures, particularly norm-based measures, grow with the size of the model and thus incorrectly predict that larger networks generalize worse than smaller networks. Other approaches have tried to establish a connection between model compression and generalization (Arora et al., 2018; Zhou et al., 2019). While both of these approaches are theoretically elegant and yield tighter bounds than bounds that are based on the size of uncompressed networks, they nonetheless grow with the size of the original network. Recent empirical studies (Jiang et al., 2020; 2019), on the other hand, identify three classes of generalization measures that do seem predictive of generalization: measures that estimate the *flatness* of local minima, the speed of the optimization, and the margin of training samples to decision boundaries. While these measures are correlated with generalization, their failure to fully explain the test performance of the model demonstrate a need for other notions of model simplicity to explain generalization in neural networks.

In this paper, we leverage the empirical observation that large fractions of trained neural networks' parameters can be *pruned* – that is, set to 0 – without hurting the models' performance (Gale et al., 2019; Zhu & Gupta, 2018; Han et al., 2015). Based on this insight, we introduce a new measure of

a model's simplicity which we call *prunability*: the smallest fraction of weights we can keep while pruning a network without hurting its training loss.

In a range of empirical studies, we demonstrate that a model's prunability is indeed highly informative of its generalization. In particular, we find that the larger the fraction of parameters that can be pruned without hurting a model's training loss, the better the model will generalize. Overall, we show that the smaller the fraction of parameters a model actually "uses" - the *simpler* it is - the better the network's generalization performance.

In summary, our main contributions are thus the following:

1. We introduce a new generalization measure called *prunability* that captures a model's simplicity (Sec. 4) and show that across a large set of models this measure is highly informative of a network's generalization performance (Sec. 5.4.1).

2. We show that even in a particularly challenging setting in which we observe a test loss double descent (Nakkiran et al., 2020; He et al., 2016), prunability is informative of models' test performance and is competitive with an existing strong generalization measures (Sec. 5.4.2).

3. Lastly, we investigate whether the success of prunability can be explained by its relationship to flat local minima. We find that while prunability makes similar predictions to some existing measures that estimate the flatness of minima, it differs from them in important ways, in particular exhibiting a stronger correlation with generalization performance than these flatness-based measures (Sec. 5.4.1 and 5.4.3).

## 2 RELATED WORK

Zhang et al. (2016) demonstrate that neural networks can perfectly memorize randomly labeled data while still attaining good test set performance on correctly labeled data, a phenomenon that complexity measures such as the VC-Dimension and Rademacher complexity fail to explain. More recently, increasing the parameter count of models has been shown to *improve* their generalization performance, yielding 'double descent' curves in which the test loss first decreases, then increases and then decreases again as the parameter count is increased (He et al., 2016; Nakkiran et al., 2020; Belkin et al., 2019). In contrast, many existing generalization measures grow monotonically with the model's size and thus fail to capture double descent, making it a particularly interesting setting to study new generalization measures (Maddox et al., 2020).

While generalization measures have been studied for a long time, Jiang et al. (2020) recently brought new empirical rigor to the field. They perform a large-scale empirical study by generating a large set of trained neural networks with a wide range of generalization gaps. Proposing a range of new evaluation criteria, they test how informative of generalization previously proposed generalization measures actually are. In this paper, we evaluate our new generalization measure in the same framework and use the strongest measures as baselines for comparison against our proposed measure, the neural network's prunability.

A common theme across many generalization measures is that they try to formalize Occam's Razor, the idea that *simpler* models generalize better than more complex models. Rissanen (1986) formalizes this principle as a model's Minimum Description length which was later applied to small neural networks by Hinton & Van Camp (1993). Similarly, Neyshabur et al. (2015; 2017) and numerous other approaches suggest that networks with smaller parameter norms generalize better. We are aware of two existing generalization measures that are based on compression or pruning. Arora et al. (2018) derive a PAC-Bayesian generalization bound based on the size of a network following a particular compression method based on noise stability. Zhou et al. (2019), on the other hand, formulate a generalization bound that is non-vacuous even for large network architectures based on the description length of a compressed network which in turn also grows with the size of the original network. While these approaches are theoretically elegant, all such measures grow with the number of parameters in a given network and thus do not capture the phenomenon that more highly overparameterized networks tend to generalize better.

Three classes of generalization measures seem to be particularly predictive of generalization. First, measures that estimate the *flatness* of local minima, either based on random perturbations of the

weights (Neyshabur et al., 2017; Dziugaite & Roy, 2017; Jiang et al., 2020; McAllester, 1999) or directly based on the curvature of the loss surface (Maddox et al., 2020). In a purely empirical study, Morcos et al. (2018) find that a model's robustness to clamping the *activations* of individual units to fixed values is predictive of its generalization performance. Second, margin-based measures that use the distance of training samples to decision boundaries (Jiang et al., 2019). Lastly, Jiang et al. (2020); Ru et al. (2020) have proposed measures that estimate the speed of optimization.

A large fraction of a network's parameters can be removed without impairing performance (Gale et al., 2019; Blalock et al., 2020); in fact, such pruning often *improves* generalization performance (Thodberg, 1991). There are many approaches to selecting parameters for removal; we focus on randomly selected parameters (random pruning) and removing the smallest-magnitude parameters (magnitude pruning), though other approaches exist in the literature (Han et al., 2015; Renda et al., 2020; Lee et al., 2019; Wang et al., 2020; Blalock et al., 2020). Random pruning is applied by the regularization method dropout (Srivastava et al., 2014), which is used to improve network generalization performance. Magnitude pruning can be used in an iterative pruning and retraining procedure (Frankle & Carbin, 2019) to identify sparse subnetworks that, for some sparsity levels, generalize better than their parent network. Both approaches are therefore natural candidates for a notion of network simplicity, which we will leverage in our proposed generalization measure. While research on dropout and the Lottery Ticket Hypothesis suggest that pruning *during training* and generalization are connected, we study the distinct question of whether pruning *after training* and generalization are connected.

## 3 NOTATION

Following the notation of Jiang et al. (2020), we denote by $\mathbb{D}$ the distribution over inputs $\mathbf{X}$ and their labels $y$ and let $\kappa$ denote the number of classes. By $\mathbb{S}$ we denote a given data set containing $m$ i.i.d. tuples $\{(\mathbf{X}_1, y_1), \ldots, (\mathbf{X}_m, y_m)\}$ drawn from $\mathbb{D}$ where $\mathbf{X}_i \in \mathcal{X}$ is the input data and $y_i \in \{1, \ldots, \kappa\}$ the class labels. We denote a feedforward neural network by $f_{\boldsymbol{w}} : \mathcal{X} \to \mathbb{Z}$, its weight parameters by $\boldsymbol{w}$ and the number of its parameters by $\omega$. We denote the weight tensor of the $i^{th}$ layer of the network by $\mathbf{W}_i$ and we then have that $\boldsymbol{w} = vec(\mathbf{W}_1, \ldots, \mathbf{W}_d)$, where $d$ is the depth of the network and $vec$ denotes the vectorization operator.

Let $L$ be the 1-0 classification error under the data distribution $\mathbb{D}$ : $L(f_{\boldsymbol{w}}) \coloneqq \mathbb{E}_{(\mathbf{X}, y) \sim \mathbb{D}}[f_{\boldsymbol{w}}(X) \neq y]$, and let $\hat{L}$ be the empirical 1-0 error on $\mathbb{S}$: $\hat{L}(f_{\boldsymbol{w}}) \coloneqq \frac{1}{m} \sum_{i=1}^{m}[f_{\boldsymbol{w}}(X_i) \neq y_i]$. We define the **generalization gap** as $g(f_{\boldsymbol{w}}) \coloneqq L(f_{\boldsymbol{w}}) - \hat{L}(f_{\boldsymbol{w}})$. When training a network, we denote each hyperparameter by $\theta_i \in \Theta_i$ for $i = 1, \ldots n$ where $n$ is the total number of hyperparameters. We denote a particular hyperparameter configuration by $\boldsymbol{\theta} \coloneqq (\theta_1, \theta_2, \ldots, \theta_n) \in \Theta$, where $\Theta \coloneqq \Theta_1 \times \Theta_2 \times \cdots \times \Theta_n$.

## 4 PRUNABILITY

A generalization measure should satisfy three straightforward desiderata: 1) It should correlate with the generalization gap across a broad range of models. 2) It should not suffer trivially from Goodhart's law, that is, incorporating it into the optimization procedure of a model should lead to improved generalization. 3) It should ideally have a plausible theoretical justification to allow for further study of its properties.

A measure based on magnitude- and random pruning presents a promising direction for satisfying all three desiderata. First, it is well-established that we can remove large fractions of a neural network's parameters without negatively impacting its train performance, while also improving its test performance (Blalock et al., 2020; Thodberg, 1991). Second, training models using dropout, i.e. random pruning, is widely known to improve generalization (Srivastava et al., 2014), as does the iterative magnitude pruning procedure of Frankle & Carbin (2019) (up to certain sparsity levels). Finally, pruning has a theoretical grounding in the minimum description length principle and in PAC-Bayesian bounds.

The connection between dropout and generalization has been studied in the PAC-Bayesian framework (McAllester, 1999), which allows us to quantify the worst-case generalization performance of random predictors. PAC-Bayesian bounds define model complexity as the KL divergence be-

tween some fixed prior distribution $P$, and the probability distribution over functions induced by the random predictor (often called the posterior $Q$, although this may not correspond to a Bayesian posterior) to obtain generalization bounds of the following form.

**Theorem 1** (McAllester (1999)). *For any $\delta > 0$, data distribution $D$, prior $P$, with probability $1 - \delta$ over the training set, for any posterior $Q$ the following bound holds:*

$$\mathbb{E}_{\boldsymbol{v} \sim Q}[L(f_{\boldsymbol{v}})] \leq \mathbb{E}_{\boldsymbol{v} \sim Q}[\hat{L}(f_{\boldsymbol{v}})] + \sqrt{\frac{KL(Q||P) + \log(\frac{m}{\delta})}{2(m-1)}}$$

McAllester (2013) shows that in the case where both the prior and posterior are distributions over networks to which dropout with a given dropout probability $\alpha$ is applied, the KL term becomes: $KL(Q||P) = \frac{1-\alpha}{2}||\boldsymbol{w}||^2$. We can directly use this formalization to derive a generalization bound for networks that are randomly pruned to the largest $\alpha$, s.t. $\mathbb{E}_{\boldsymbol{v} \sim Q}[\hat{L}(f_{\boldsymbol{v}})] < \hat{L}(f_{\boldsymbol{w}}) \times (1 + \beta)$, i.e., such that the training loss does not increase more than a relative tolerance $\beta$. We have to adjust the bound to account for the search for $\alpha$. We make use of the fact that we search over a fixed number $c$ of possible values for $\alpha$ and use union bound which will change the log term in the bound to $\log(\frac{cm}{\delta})$. We thus obtain the following bound:

$$\mathbb{E}_{\boldsymbol{v} \sim Q}[L(f_{\boldsymbol{v}})] \leq \mathbb{E}_{\boldsymbol{v} \sim Q}[\hat{L}(f_{\boldsymbol{v}})] + \sqrt{\frac{\frac{1-\alpha}{2}||\boldsymbol{w}||_2^2 + \log(\frac{m}{\delta}) + 5}{2(m-1)}} \ . \tag{1}$$

While this bound will be vacuous for many large neural networks, it does provide us with some intuition for why we should expect a model's generalization to be linked to its robustness to pruning. In particular, this bound predicts the *relative* generalization performance of models from which different fractions of parameters can be removed without adversely affecting their training loss. It is this prediction that we evaluate with the experiments described below. We provide the details on the derivation in Appendix C.

Given a trained model and a pruning algorithm, we define the model's **prunability** as the smallest fraction of the model's parameters that we can keep while pruning without increasing the model's training loss by more than a given relative tolerance $\beta$. We choose the *fraction* of parameters pruned rather than the *number* of parameters remaining based on the empirical observation that naive parameter counting is a poor predictor of generalization performance.

**Definition 1** (Prunability). *Let $\hat{L}$ denote the cross-entropy loss. For a given model $f_{\boldsymbol{w}}$, a training data set $\mathbb{S}_{train}$, a relative bound on the change in train loss $\beta$ and a pruning method $\phi$, we thus define prunability as*

$$\begin{aligned} \mu_{prunability}(f_{\boldsymbol{w}}) = \quad & \arg\min_{\alpha} \phi(f_{\boldsymbol{w}}, \alpha) \\ s.t. \quad & \hat{L}(\phi(f_{\boldsymbol{w}}, \alpha), \mathbb{S}_{train}) \leq (1 + \beta) \times \hat{L}(f_{\boldsymbol{w}}, \mathbb{S}_{train}) \ , \end{aligned} \tag{2}$$

*where $\phi(f_{\boldsymbol{w}}, \alpha)$ sets the fraction $1 - \alpha$ of the weights of $f_{\boldsymbol{w}}$ to zero.*

In our experiments we evaluate robustness to pruning both with regards to **magnitude pruning**, in which the parameters with the smallest absolute magnitude are set to 0, and **random pruning**, where random parameters are set to 0. Magnitude pruning can remove a greater number of model weights without affecting performance than random pruning (Blalock et al., 2020), and can therefore be considered a more effective pruning strategy. We hypothesize that using a more effective pruning method might yield a measure of prunability that is more predictive of generalization. Per default, we refer to a model's prunability with regards to magnitude pruning in the following. Note that while *iterative* pruning is usually used when the aim is to achieve the best compression rate, we are only concerned with the generalization of the model at the end of training and thus apply one-shot pruning. In our experiments we use a tolerance $\beta$ of 0.1 and note that the particular choice of $\beta$ did not seem to have a large impact on the predictiveness of prunability. See Appendix A.6.1 for pseudo-code of how prunability is computed for a given model.

## 5 EXPERIMENTAL RESULTS AND ANALYSIS

The goals of our experiments are fourfold: 1) We verify that prunability predicts generalization across a set of models trained with different hyperparameters and generalization performance. To

this end, we use a similar experimental framework as Jiang et al. (2020). 2) We evaluate how prunability performs in a particularly challenging double descent setting. 3) To understand what notion of simplicity prunability quantifies, we study the conditional mutual information between it and baselines that represent existing classes of different strong generalization measures. 4) Given that prunability and random perturbation robustness both measure the sensitivity of a model's training loss to a perturbation of its weights, we directly compare how pruning and random weight perturbations impact models' losses.

## 5.1 BASELINES

We compare our generalization measure to a number of baseline measures. Having somewhat tighter computational constraints than Jiang et al. (2020), we select one strong baseline per class of generalization measures as described by Jiang et al. (2020) and Jiang et al. (2019).

From the class of generalization measures that aim to capture a model's simplicity using parameter-norms, we use the **Sum of Squared Frobenius Norms** (Jiang et al., 2020):

$$\mu_{\text{sum\_of\_fro}}(f_{\boldsymbol{w}}) = d \cdot (\prod_{i=1}^{d} ||\mathbf{W}_i||_F^2)^{1/d},$$

where $d$ is the number of layers of the model. From the class of measures based on the idea that models with flatter local minima generalize better, we use **magnitude aware random perturbation robustness** (Jiang et al., 2020):

$$\mu_{\text{pac\_bayes\_mag\_flat}}(f_{\boldsymbol{w}}) = \frac{1}{\sigma^2},$$

where $u_i \sim \mathcal{N}(0, \sigma^2|w_i| + \epsilon)$ and $\sigma^2$ is the largest number such that $\mathbb{E}_u[\hat{L}(f_{\boldsymbol{w}+\boldsymbol{u}})] \leq (1+\beta) \times \hat{L}(f_{\boldsymbol{w}})$, where $\beta$ is an arbitrarily selected relative tolerance, in our experiments 0.1. From the class of measures based on the idea that models with wider margins between the training samples and decision boundaries generalize better, we use the **Normalized Margins** measure (Jiang et al., 2019):

$$\mu_{\text{norm\_margins}}(f_{\boldsymbol{w}}) = \boldsymbol{a}^T \boldsymbol{\phi},$$

wherein $\boldsymbol{\phi}$ consists of five statistics of the normalized margins distribution for each layer of the model. That is, at each hidden layer of the network, the normalized distances of all samples to the decision boundary of the second most likely class are computed. Five statistics of each of these distance distributions are used as input for the linear regression. Lastly, from the class of measures based on the idea that the speed of optimization is predictive of generalization, we use the negative **Sum of Train Losses** (Ru et al., 2020) throughout training:

$$\mu_{\text{SoTL}}(f_{\boldsymbol{w}}) = -\sum_{t=1}^{T} \hat{L}(f_{w^t}),$$

where $T$ is the number of training epochs, $w^t$ the parameters at epoch $t$ and $\hat{L}$ the training cross-entropy loss. In Appendix A.2 we provide further details on these generalization measures and additional baselines.

## 5.2 MODELS

To generate a large set of trained models, we build upon the DEMOGEN set of trained models introduced by Jiang et al. (2019). Our data set consists of 324 models that resemble the Network-in-Network architecture by (Lin et al., 2014) trained on CIFAR10. Jiang et al. (2019) propose this particular architecture because it is parameter-efficient and achieves relatively competitive performance on standard image classification benchmarks. We achieve a wide range of generalization gaps by varying a number of hyperparameters such as the width, depth or $L_2$-regularization of the models. We modify the original set of models, DEMOGEN, used by Jiang et al. (2019) to obtain an even stronger experimental set up: noticing that the *depth* of neural networks has a large impact on generalization performance, we add models of different depths to the original DEMOGEN data set. Following Jiang et al. (2020) we only use models that apply batch normalization. See Appendix A.3 for further details of our experimental set-up.

## 5.3 EVALUATION METRICS

To study our generalization measures comprehensively, we consider 1) their Kendall's Rank correlation with the generalization gap, 2) how predictive they are in terms of the adjusted $R^2$ and 3) apply the conditional independence test of Jiang et al. (2020), in an attempt to understand the causal relations between the complexity measures and generalization. We briefly summarize the metrics in this section and provide a more detailed explanation in Appendix A.1

We compute **Kendall's rank correlation coefficient** which quantifies to what extent the ranking of models according to a given generalization measure corresponds to the ranking according to the generalization gaps. For a given generalization measure $\mu$, we consider: $\mathcal{T} := \cup_{\boldsymbol{\theta} \in \Theta}\{(\mu(\boldsymbol{\theta}), g(\boldsymbol{\theta}))\}$ where $g(\boldsymbol{\theta})$ is the generalization gap of the model trained with hyperparameters $\boldsymbol{\theta}$. Kendall's rank correlation coefficient is then defined as the fraction of pairs of tuples that are correctly ranked according to the generalization measure: $\tau(\mathcal{T}) := \frac{1}{|\mathcal{T}|(|\mathcal{T}|-1)} \sum_{(\mu_1, g_1) \in \mathcal{T}} \sum_{(\mu_2, g_2) \in \mathcal{T} \setminus (\mu_1, g_1)} \text{sign}(\mu_1 - \mu_2)\text{sign}(g_1 - g_2)$. While Kendall's rank correlation coefficient is generally a useful metric, it does not adequately reflect whether a generalization measure's performance is consistent across all hyperparameter axes. Thus, we also consider the **granulated Kendall's rank correlation coefficient** which is essentially the average of Kendall's rank coefficients $\psi_i$ obtained by only varying one hyperparameter dimension at a time: $\Psi := \frac{1}{n} \sum_{i=1}^{n} \psi_i$

To go beyond correlation, Jiang et al. (2020) use a conditional independence test inspired by Verma & Pearl (1991). The main goal here is to understand whether there exists an edge in a causal graph between a given generalization measure $\mu$ and the generalization gap $g$: this tells us whether models generalize well *because* of the value of the generalization measure, or whether the two values are only correlated. This is achieved by estimating the **conditional mutual information** between the generalization measure and the generalization gap conditioned on different hyperparameters that are observed. Formally, we have that for any function $\phi : \Theta \to \mathcal{R}$, let $V_\phi : \Theta_1 \times \Theta_2 \to \{+1, -1\}$ be as $V_\phi(\theta_1, \theta_2) := sign(\phi(\theta_1) - \phi(\theta_2))$. Let $U_{\mathcal{S}}$ be a random variable that corresponds to the values of the hyperparameters in $\mathcal{S}$. Intuitively, the higher this metric is, the more likely it is that there is indeed an edge in the causal graph between the generalization gap and the generalization measure: $\mathcal{K}(\mu) = \min_{U_{\mathcal{S}} s.t. |\mathcal{S}| \leq 2} \hat{\mathcal{I}}(V_\mu, V_g | U_{\mathcal{S}})$, where $\hat{\mathcal{I}}(V_\mu, V_g | U_{\mathcal{S}})$ is the normalized conditional mutual information between the generalization measure and the generalization gap conditioned on observing the hyperparameter set $\mathcal{S}$.

To evaluate how well we can predict a model's generalization using a linear function of a given generalization measure and following Jiang et al. (2019), we additionally evaluate the generalization measures' predictiveness in terms of their adjusted $R^2$, i.e. the proportion of the variance of the generalization gap $g(\boldsymbol{\theta})$ across hyperparameters $\boldsymbol{\theta} \in \Theta$ that can be explained by the generalization measure $\mu(\boldsymbol{\theta})$.

## 5.4 RESULTS

### 5.4.1 PERFORMANCE OF PRUNABILITY ON EXTENDED DEMOGEN DATA SET

Table 1 holds a summary of our experimental results on the extended DEMOGEN data set. We find that prunability is highly informative of generalization across all of our evaluation metrics. In particular, it outperforms random perturbation robustness, the training loss itself and the Frobenius norm measures in terms of Kendall's rank correlation coefficient.

We find that while prunability based on magnitude pruning outperforms prunability based on random pruning, the difference in performance is relatively small.

We find that the Normalized Margins baseline clearly outperforms prunability while the Negative Sum of Train Losses and the Best Margin Variable seem similarly informative of generalization as prunability. Note that and the margin-based measures achieve much higher conditional mutual information scores than the other baselines, providing further evidence for the causal connection between wide margins and good generalization performance. The conditional mutual information between *prunability* and generalization is relatively low, suggesting that it seems somewhat less likely that there is a direct causal connection between a model's prunability and its generalization performance.

| Generalization Measure | Kendall's $\tau$ | Adjusted R2 | Cond. Mutual Information |
|---|---|---|---|
| Prunability (ours) | 0.6496 | 0.5440 | 0.0324 |
| Random Prunability (ours) | 0.584 | 0.3015 | 0.0377 |
| Random Perturbation Robustness | 0.4253 | 0.2440 | 0.0230 |
| Frobenius-norm | -0.6055 | 0.7778 | 0.0612 |
| Normalized Margins | **0.8061** | **0.8866** | **0.1761** |
| Best Margin Variable | 0.6344 | 0.6623 | 0.0743 |
| Negative Sum of Train Losses | 0.7085 | 0.4923 | 0.0186 |
| Train loss | 0.0201 | 0.0746 | 0.0004 |

Table 1: **Prunability is highly predictive of generalization.** Comparison of generalization measures' correlation with test performance on a set of convolutional networks trained on CIFAR-10. Higher values are better across all metrics. The standard error of the Kendall's $\tau$ is $s_\tau = 0.037$.

In Table 2, we compare the granulated and regular Kendall's coefficients of some additional baselines. The results reconfirm the weak performance of norm and parameter count-based complexity measures both for pruned and unpruned networks. While this a well known issue, we point out that this is also problematic for recently proposed generalization measures based on the size of the compressed network such as Zhou et al. (2019). We also note that the Frobenius norm measure is strongly negatively correlated with generalization rather than positively correlated as one would expect based on Jiang et al. (2020). It seems that this result is largely due to a strong negative correlation for models with different dropout rates and relatively weak correlation coefficients along other hyperparameter axes.

| Generalization Measure | Width | Dropout Rate | Data Augment. | Weight Decay | Depth | Kendall $\tau$ | $\psi$ |
|---|---|---|---|---|---|---|---|
| prunability (ours) | **0.1582** | 0.6986 | 0.0122 | -0.001 | 0.2254 | 0.6496 | 0.2187 |
| random_prunability (ours) | 0.0714 | 0.6261 | 0.0464 | 0.0422 | 0.259 | 0.584 | 0.209 |
| random_perturbation | 0.1161 | 0.5154 | 0.0962 | 0.0755 | 0.1829 | 0.4253 | 0.1972 |
| frobenius_norm | -0.0778 | -0.7955 | -0.067 | -0.0591 | 0.4577 | -0.6055 | -0.1083 |
| normalized_margins | 0.0671 | **0.8325** | **0.1121** | 0.0799 | **0.5789** | **0.8061** | **0.3341** |
| best_margins_variable | 0.054 | 0.7756 | 0.0415 | -0.039 | 0.4798 | 0.6344 | 0.2624 |
| sum_of_train_losses | -0.0639 | 0.8078 | 0.0498 | 0.0345 | 0.4156 | 0.7085 | 0.2487 |
| train_loss | 0.0143 | -0.0652 | -0.0762 | -0.0349 | 0.1142 | 0.0201 | -0.0096 |
| sum_two_norms | -0.0513 | -0.5728 | 0.0214 | 0.0746 | -0.1359 | -0.2096 | -0.1328 |
| parameter_count | -0.0866 | 0.0 | 0.0 | 0.0 | -0.4218 | -0.0911 | -0.1017 |
| sum_two_norms_pruned | -0.0749 | -0.6828 | 0.0857 | **0.0866** | -0.2654 | -0.2234 | -0.1701 |
| pruned_parameter_count | -0.0893 | 0.6986 | 0.0122 | -0.001 | -0.3798 | 0.0417 | 0.0481 |

Table 2: **Prunability achieves a stronger granulated rank correlation than measures based on random perturbation robustness or norms.** Columns labeled with hyperparameters (width, dropout rate, etc.) indicate Kendall $\tau$ if we only vary this hyperparameter. The last two columns are the Overall Kendall's $\tau$ and the Granulated Kendall's coefficient $\psi$, which is the average of the per-hyperparameter columns. Higher values are better across all columns

### 5.4.2 DOUBLE DESCENT

In addition to our experiments on the DEMOGEN data set, we also evaluate prunability on a set of models that exhibit a test loss double descent (Nakkiran et al., 2020). This phenomenon is particularly interesting for the study of generalization measures, since many measures that aim to capture a model's complexity grow monotonically as we increase a network's width and thus do not display a double descent. We are thus including this experiment to evaluate whether prunability also succumbs to this phenomenon. We use the experimental setup proposed by Maddox et al. (2020), and compare our new measure to their curvature-based generalization measure which also aims to capture double descent. Note that up to now we studied the generalization gap in terms of the $0 - 1$ loss. In this section, however, we focus on another aspect of generalization and study the relation between prunability and models' cross-entropy test loss.

Maddox et al. (2020) introduce a generalization measure called Effective Dimensionality which is defined as: $N_{\text{eff}}(\mathbf{H}, z) = \sum_{i=1}^{k} \frac{\lambda_i}{\lambda_i + z}$ where $\lambda_i$ are the eigenvalues of the Hessian of the training loss $\mathbf{H} \in \mathcal{R}^{k \times k}$, and $z > 0$ is a regularization constant. Intuitively, the Effective Dimensionality measures the flatness of the local minimum of a network in terms of the number of eigenvalues of the Hessian of the training loss that are "large". To get a measure on the same scale as our

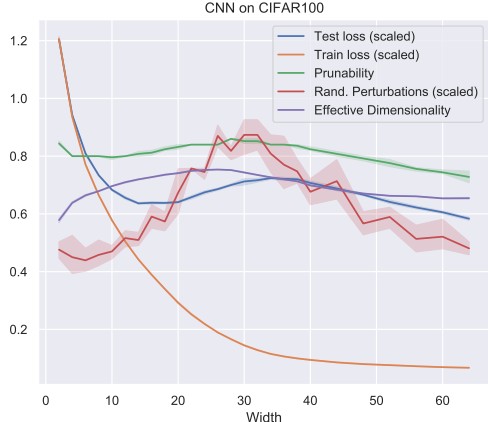

| Kendall's $\tau$ | Test Loss | Test Error |
|---|---|---|
| Prunability | **0.3613** | 0.1277 |
| Eff. Dim. | -0.0197 | -0.0753 |
| Random Perturbation | 0.1398 | -0.2166 |
| Frobenius Norm | 0.0996 | -0.5367 |
| Margins | 0.1296 | **0.4258** |
| -Sum of Train Losses | -0.2890 | -0.9381 |

(a) Comparison of prunability and baseline measures in a double descent setting.

(b) Kendall's rank correlation ($s_\tau = 0.06$) between generalization measures and test loss. Higher is better.

Figure 1: **Prunability is correlated with test loss in double descent setting:** Across a set of convolutional networks of varying width trained on CIFAR-100, we show that prunability captures double descent behavior better than a previously proposed metric *Effective Dimensionality* – which is based on the eigenspectrum of the Hessian of the training loss of the model – and other strong baselines.

new generalization measure, we use a normalized version of $N_{eff}$: $\mu_{\text{effective\_dimensionality}} = \frac{N_{\text{eff}}}{k}$. Additionally, we also evaluate a number of the strongest baselines from Section 5.4.1 in the double descent setting. We refer to appendix A.4 for further details on the experimental setup.

We find that prunability does indeed display a double descent and is more strongly rank correlated with the test loss than the baseline measures (Kendall's $\tau$ of 0.3613, $p < 0.0005$).

In Appendix B.1 we study an additional double descent setting and obtain results that are consistent with the results presented in this section.

### 5.4.3 PRUNING AND FLATNESS-BASED MEASURES

We now investigate whether a network's robustness to pruning can be explained by other established generalization measures, or whether it is measuring something entirely distinct from previous notions of model simplicity. To better understand the relation between prunability and the other generalization measures we perform the conditional independence test described in section A.1.3 between them. Recall that, up to now, we used this test to study the relation between the generalization measure and the generalization gap and that the higher the $\kappa$ between two variables, the likelier it is that they neighbors in the causal DAG. See Table 3 for a summary of the results.

First, we observe that the conditional mutual information between prunability and random perturbation robustness is the highest across the baselines we are evaluating. Because pruning is a special case of perturbation, some connection between the two methods is expected. However, prunability is clearly measuring something subtly different from flatness of the loss surface, as it outperforms random perturbation robustness in our evaluations. Thus, we investigate whether this distinction may arise from the differential effects of pruning and random perturbations in function space.

To evaluate this, we directly compare the impact of both pruning and random perturbations on some of the models in our double descent experimental setting by performing pruning and random perturbations of the same magnitude and computing their respective impact on the loss. We find that pruning has a larger negative impact on a model's training loss than a random perturbation of the same magnitude (see Figure 2). Pruning can lead to an improvement of the test loss which we do not observe for random perturbations; we conjecture that this relationship may be responsible for the difference between the two methods, though disentangling this phenomenon is left for future work. We also note that prunability achieves much higher Kendall's $\tau$, Adjusted $R^2$ and Conditional Mutual Information than random perturbation robustness.

In conclusion, pruning affects models differently from randomly perturbing their weights. Because pruning can in certain settings improve test performance (and hence seems to be more aligned with network simplicity) we suggest that this difference leads to the stronger performance of prunability as compared to random perturbation robustness. We compare the impact of pruning and random perturbations on a model's training and test loss for an additional architecture in Appendix B.2, and find that the results are consistent with the results presented in this section.

|  | Normalized Margins | Random Perturbations | Parameter Count | Train Loss | Frobenius Norm | SoTL |
|---|---|---|---|---|---|---|
| Prunability | 0.0122 | 0.04541 | 0.0038 | 0.0004 | 0.0105 | 0.0013 |

Table 3: **Prunability is similar to Random Perturbation Robustness.** The conditional mutual information (CMI) between prunability and random perturbation robustness.

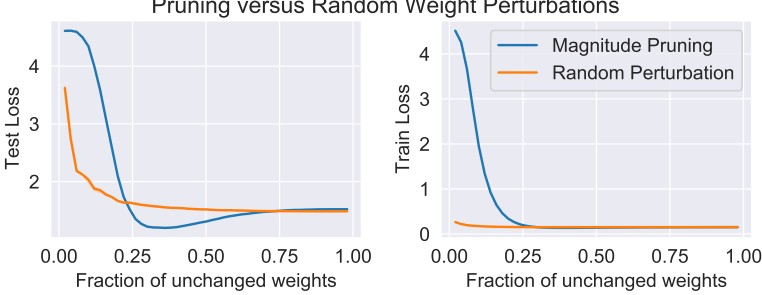

Figure 2: **Pruning affects models very differently than random perturbations.** Here we compare pruning of weights and randomly perturbing the same weights by the same amount. We study a ResNet18 trained on CIFAR100. Generally speaking, pruning will have a larger negative impact on a model's loss than randomly perturbing the same weights by the same amount but in some cases pruning actually improves the test loss of models.

## 6 CONCLUSION

In this paper, we show that a network's prunability, the smallest fraction of parameters we can keep while pruning without hurting the model's training loss, is highly informative of the network's generalization performance. We also show that prunability seems to capture something distinct from previously suggested measures based on robustness to random perturbations or the curvature of the loss landscape. We see a number of promising directions for further research. For instance, one could study whether the direction of pruning lies in the span of the main eigenvectors of the Hessian studied by Maddox et al. (2020). Given the strong empirical performance of prunability – and given that the theory behind its success is not yet well understood – this measure of model complexity may be of use to the construction of future generalization bounds. We include an attempt of this in Appendix C, but more thorough investigation will likely prove illuminating.

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

# A    EXPERIMENTAL SETUP

## A.1    EVALUATION METRICS

In this section we describe the metrics we use to evaluate the generalization measures. To study our generalization measures comprehensively, we consider the generalization measures' Kendall's Rank correlation with the generalization gap, how predictive they are in terms of Adjusted $R^2$ and Jiang et al. (2020).'s conditional independence test in an attempt to understand the causal relations between the complexity measures and generalization.

### A.1.1    KENDALL'S RANK CORRELATION COEFFICIENT

This criterion evaluates the quality of a given generalization in terms of how well it *ranks* models with different generalization gaps. Intuitively, a generalization measure ranks two models that were trained with hyperparameters $\boldsymbol{\theta}_1$ and $\boldsymbol{\theta}_2$ correctly iff $\mu(\boldsymbol{\theta}_1) > \mu(\boldsymbol{\theta}_2) \rightarrow g(\boldsymbol{\theta}_1) > g(\boldsymbol{\theta}_2)$ is satisfied. Kendall's rank coefficient then measures for what fraction of models in our data set this holds. Kendall T takes values between 1 (perfect agreement) and -1 (rankings are exactly reversed) where a Kendall's t of 0 means that the complexity measure and generalization gap are independent.

Formally, we consider the set of tuples of generalization measures and generalization gaps:

$$\mathcal{T} := \cup_{\boldsymbol{\theta} \in \Theta}\{(\mu(\boldsymbol{\theta}), g(\boldsymbol{\theta}))\}$$

And Kendall's rank correlation coefficient is defined as:

$$\tau(\mathcal{T}) := \frac{1}{|\mathcal{T}|(|\mathcal{T}|-1)} \sum_{(\mu_1, g_1) \in \mathcal{T}} \sum_{(\mu_2, g_2) \in \mathcal{T} \setminus (\mu_1, g_1)} sign(\mu_1 - \mu_2)sign(g_1 - g_2)$$

### A.1.2    GRANULATED KENDALL'S RANK COEFFICIENT

While Kendall's rank correlation coefficient is generally a useful metric, it does not adequately reflect whether a generalization measure's performance is consistent across all hyperparameter axes. To account for this Jiang et al. (2020) propose the granulated Kendall's rank-correlation coefficient which is essentially the average of Kendall's rank coefficients obtained by only varying one hyperparameter dimension at a time.

Formally, we have:

$$m_i := |\Theta_1 \times \cdots \times \Theta_{i-1} \times \Theta_{i+1} \times ...\Theta_n|$$

and the Kendall's rank correlation coefficient is defined as:

$$\psi_i := \frac{1}{m_i} \sum_{\theta_1 \in \Theta_1} \cdots \sum_{\theta_{i-1} \in \Theta_{i-1}} \sum_{\theta_{i+1} \in \Theta_{i+1}} \cdots \sum_{\theta_n \in \Theta_n} \tau(\cup_{\theta_i \in \Theta_i}(\mu(\boldsymbol{\theta}), g(\boldsymbol{\theta})))$$

Averaging all of these values, we obtain the granulated Kendall's rank correlation coefficient:

$$\Psi := \frac{1}{n} \sum_{i=1}^{n} \psi_i$$

Given these definitions, we obtain the following standard errors for the granulated Kendall's coefficients in our experiments. For the width, dropout rate, weight decay and depth measurements we hsve, $s_\tau = 0.00056$ and for the data augmentation measurements we have $s_\tau 0.000484$. Lastly, for $\psi$, we have $s_\tau = 0.00011$

### A.1.3 CONDITIONAL MUTUAL INFORMATION

To go beyond measures of correlation, Jiang et al. (2020) use a conditional independence test inspired by Verma & Pearl (1991). The main goal here is to understand whether there exists an edge in a causal graph between a given generalization measure $\mu$ and the generalization gap $g$. This is achieved by estimating the mutual information between the generalization measure and the generalization gap conditioned on different hyperparameters being observed. The higher this metric is, the more likely it is that there is indeed an edge in the causal DAG between the generalization gap and the generalization measure.

Formally, we have that for any function $\phi : \Theta \to \mathcal{R}$, let $V_\phi : \Theta_1 \times \Theta_2 \to \{+1, -1\}$ be as $V_\phi(\theta_1, \theta_2) \coloneqq sign(\phi(\theta_1) - \phi(\theta_2))$. Let $U_\mathcal{S}$ be a random variable that corresponds to the values of the hyperparameters in $\mathcal{S}$. The conditional mutual information can then be computed as:

$$\mathcal{I}(V_\mu, V_g | U_\mathcal{S}) = \sum_{U_\mathcal{S}} p(U_\mathcal{S}) \sum_{V_\mu \in \{\pm 1\}} \sum_{V_g \in \{\pm 1\}} p(V_\mu, V_g | U_\mathcal{S}) \log \left( \frac{p(V_\mu, V_g | U_\mathcal{S})}{p(V_\mu | U_\mathcal{S}) p(V_g | U_\mathcal{S})} \right)$$

By dividing the term above by the conditional entropy of generalization:

$$\mathcal{H}(V_g | U_\mathcal{S}) = -\sum_{U_\mathcal{S}} p(U_\mathcal{S}) \sum_{V_\mu \in \{\pm 1\}} p(V_g | U_\mathcal{S}) \log(p(V_g | U_\mathcal{S}))$$

we obtain the normalized criterion:

$$\hat{\mathcal{I}}(V_\mu, V_g | U_\mathcal{S}) = \frac{\mathcal{I}(V_\mu, V_g | U_\mathcal{S})}{\mathcal{H}(V_g | U_\mathcal{S})}$$

In the original Inductive Causation algorithm (Verma & Pearl, 1991), an edge in the causal graph is kept if there is no subset of hyperparameters $\mathcal{S}$ such that the two two nodes are independent, that is $\hat{\mathcal{I}}(V_\mu, V_g | U_\mathcal{S}) = 0$. Due to computational limitations, and following Jiang et al. (2020), we only consider subsets of hyperparameters of size at most 2:

$$\mathcal{K}(\mu) = \min_{U_\mathcal{S} s.t. |\mathcal{S}| \leq 2} \hat{\mathcal{I}}(V_\mu, V_g | U_\mathcal{S})$$

Intuitively, the higher $\mathcal{K}(\mu)$, the more likely it is that there exists an edge between $\mu$ and generalization in the causal graph.

### A.1.4 ADJUSTED $R^2$

Lastly, we consider a metric to evaluate how predictive a generalization measure is of generalization based on a simple linear regression: $\hat{g} = \boldsymbol{a}^T \mu + b$. We find $\boldsymbol{a}$ and $b$ by minimizing the mean squared error: $a^*, b^* = \arg\min_{\boldsymbol{a},b} \sum_i (\boldsymbol{a}^T \phi(\boldsymbol{\mu}_i) + b - g_i)^2$

On a test set of models we then compute the coefficient of determination $R^2$ which measures what fraction of the variance of the data ca be explained by the linear model:

$$R^2 = 1 - \frac{\sum_{j=1}^n (\hat{g}_j - g_j)^2}{\sum_{j=1}^n (g_j - \frac{1}{n} \sum_{j=1}^n g_j)^2}$$

We compute the $R^2$ using a 10-fold cross-validation.

Alternatively, one can use the adjusted $R^2$ score ($\bar{R}^2$) which measures how well a linear model fits a training set and which accounts for the number of variables used in the regression:

$$\bar{R}^2 = 1 - (1 - R^2) \frac{n - 1}{n - dim(\boldsymbol{\mu}) - 1}$$

| Layer Id | Layer Type |
|----------|------------|
| 0 | 3 x 3 convolution, stride 2 |
| 1 | 1 x 1 convolution, stride 1 |
| 2 | 1 x 1 convolution, stride 1 |

Table 4: A NiN block

## A.2 GENERALIZATION MEASURES

Additional generalization measures we consider are:

- $\mu_{\text{best\_margins\_variable}}$: To account for the fact that the normalized margins variable is based on a large number of input variables, we select the single independent variable that has the largest standardized coefficient in the Normalized Margins model and directly use it as a generalization measure.

- $\mu_{\text{train\_loss}}$: $\hat{L}(f_{\boldsymbol{w}})$, the cross entropy loss of the model $f_{\boldsymbol{w}}$ on the training data set after training is finished.

- $\mu_{\text{parameter\_count}}$: The number of parameters $\omega$ of the model.

- $\mu_{\text{sum\_two\_norms}}$:

$$\mu_{\text{sum\_of\_two\_norms}}(f_{\boldsymbol{w}}) = (\sum_{i=1}^{d} ||\mathbf{W}_i||_2^2)$$

- $\mu_{\text{sum\_two\_norms\_pruned}}$: Essentially, the same as $\mu_{\text{sum\_two\_norms}}$, but applied to $f_{\boldsymbol{w}}$ that we obtain when we compute $\mu_{prunability}$. Formally, we have:

$$\mu_{\text{sum\_two\_norms\_pruned}}(f_{\boldsymbol{w}}) = \mu_{\text{sum\_two\_norms}}(\min_{\alpha} \phi(f_{\boldsymbol{w}}, \alpha))$$

$$\text{s.t.} \quad \hat{L}(\phi(f_{\boldsymbol{w}}, \alpha), \mathbb{S}_{train}) \leq (1 + \beta) \times \hat{L}(f_w, \mathbb{S}_{train})$$

- Analogously, $\mu_{\text{pruned\_parameter\_count}}$ is the parameter count $\omega$ of the pruned model we obtain while computing $\mu_{\text{prunability}}$

## A.3 DEMOGEN

We extend the DEMOGEN data set in Jiang et al. (2019) with regards to two aspects. First, we follow Jiang et al. (2020) and only consider models that are trained using batch normalization. Second, Jiang et al. (2019) do not consider networks of different depths. We consider this to be a crucial hyperparameter and a extend the original DEMOGEN data set by training additional models with different numbers of NiN-blocks than the models in the original data set.

We use the network architecture proposed by Jiang et al. (2019) in the DEMOGEN data set which is very similar to the architecture used in Jiang et al. (2020). The architecture is very similar to the Network in Network architecture in Lin et al. (2014) with the max pooling and dropout layer removed. We train the networks on CIFAR-10 (Krizhevsky et al., 2009). The architecture we use is thus:

1. The input layer
2. 2, 3 or 4 NiN-blocks as described in Table 4.
3. A single convolution layer with kernel size 4x4 and stride 1.

In this architecture we vary the following hyperparameters:

1. Use channel sizes of 192, 288, and 384
2. Apply dropout after NiN-blocks with $p = 0.0, 0.2, 0.5$
3. Apply $l_2$ regularization with $\lambda = 0.0, 0.001, 0.005$.
4. Train with and without data augmentation, that is random cropping, flipping and shifting.

5. We consider two random initializations per hyperparameter configuration.

6. Models with 2, 3 or 4 NiN blocks as described above.

In total we are thus working with a data set of 324 trained models. We train the models with SGD with momentum ($\alpha = 0.9$) with a batch size of 128 and an initial learning rate of 0.01. The networks are trained for 380 epochs with a learning rate decay of 10x every 100 epochs. We base our implementation on the original code base: `https://github.com/google-research/google-research/tree/master/demogen`.

### A.4 DOUBLE DESCENT

For our double descent experiments we use the models and baselines used by Maddox et al. (2020) which are available on: `https://github.com/g-benton/hessian-eff-dim`. In particular, the CNN we use is the architecture used by Nakkiran et al. (2020) which is availbable on: `https://gitlab.com/harvard-machine-learning/double-descent/-/blob/master/models/mcnn.py` Likewise, the ResNet18 architecture can be found here: `https://github.com/g-benton/hessian-eff-dim/blob/temp/hess/nets/resnet.py`.

The other baselines correspond to the ones from the DEMOGEN experiments described in Appendix A.3. The margins baseline used in this setting is based on the one used in Jiang et al. (2019): a linear model is trained to predict the test loss based on the statistics of the margin distribution at the *output* layer of a given model.

We train a set of 32 networks of widths 2, 4, 6, ..., 64 on CIFAR100 (Krizhevsky et al., 2009) using SGD with a learning rate of $10^{-2}$, momentum of 0.9, weight decay of $10^{-4}$ for 200 epochs with a batch size of 128. The learning rate decays to $10^{-4}$ on a piecewise constant learning rate schedule (Izmailov et al., 2018), beginning to decay on epoch 100. Random cropping and flipping are used for data augmentation which is turned off for the computation of eigenvalues. Our plots contain the results across 5 random seeds. The shaded areas indicate the standard error. All experiments were run on Nvidia GeForce RTX 2080 Ti GPUs.

### A.5 RANDOM PERTURBATIONS VERSUS PRUNING

We use the ResNet18 implementation from `https://gitlab.com/harvard-machine-learning/double-descent/-/blob/master/models/resnet18k.py` and use the same hyperparameters for training as in the double descent settings above. For every magnitude pruned model we also compute the randomly perturbed model in which we perturb the same set of weights by a random vector of the same size.

### A.6 ALGORITHMS

#### A.6.1 ALGORITHM TO DETERMINE PRUNABILITY

---

**Algorithm 1:** Maximal Lossless Magnitude Pruning

---

**Input:** Original model $f_{\boldsymbol{w}}$, train_data, tolerance in change of loss $\beta$, step size for search step_size

**Result:** Pruned Model $f_{\boldsymbol{w}_{pruned}}$

1   original_train_loss = loss(h, train_data);
2   fraction_of_weights_to_remove = 0.98;
3   $\boldsymbol{w}_{pruned}$ = prune($\boldsymbol{w}$, $fraction\_of\_weights\_to\_remove$);
4   **while** *loss($f_{\boldsymbol{w}_{pruned}}$, train_data)* $> (1 + \beta) \times$ original_train_loss **do**
5      $\boldsymbol{w}_{pruned}$ = prune($\boldsymbol{w}$, $fraction\_of\_weights\_to\_remove$);
6      fraction_of_weights_to_remove -= step_size;
7   **end**
8   **return** $f_{\boldsymbol{w}_{pruned}}$

---

Where prune($\boldsymbol{w}$, sparsity) is an algorithm that sets *sparsity* % of $\boldsymbol{w}$ to 0 and then returns the new vector. In our experiments we evaluate both *magnitude pruning*, where the parameters with the

smallest absolute magnitude are set to 0, and *random pruning*, where random parameters are set to 0.

Empirically, we find that the particular choice for the tolerance $\beta$ does not make a big difference with regards to predictiveness of prunability. In our experiments we use $\beta = 0.1$.

Note that to determine a model's prunability we use the differentiable cross-entropy loss rather than the 1-0 loss used elsewhere in the paper.

## B  ADDITIONAL EXPERIMENTS

### B.1  DOUBLE DESCENT

In Section 5.4.2, we conduct an experiment that suggests that prunability captures the particularly challenging double descent phenomenon. In this section, we provide an extended version of Figure 1, containing all baselines (see Figure 3a). Furthermore, we extend that experiment to an additional architecture, ResNet18 trained on CIFAR100 (see Figure 4). The results of this additional experiment are consistent with the one in the main part of the body: prunability is strongly Kendall rank correlated with the test loss and the test error. In this additional setting, however, the margins-based method outperforms prunability. We describe the details of the architecture and training of this experiment in Appendix A.5.

### B.2  PRUNING VERSUS RANDOM PERTURBATIONS

Likewise, we study the impact of pruning and random perturbations on the training and test loss in an additional experimental setting. As in the experiment in Section 5.4.3, we find that pruning and randomly perturbing weights (by a vector of the same magnitude), have very different impacts on the training and test loss (see Figure 5). In this experiment, we study the CNN used in our double descent experiment of Section 5.4.2. In particular, we find that pruning generally has a larger negative impact on the training loss than a random weight perturbation of the same magnitude. With regards to the test loss, we find that pruning has a smaller negative impact than random perturbations for moderate amounts of pruning, and a larger negative impact for larger amounts of pruning/perturbation. This matches the behavior of the first experiment. In this additional experiment we do not observe an improvement of the test loss through pruning which was not expected.

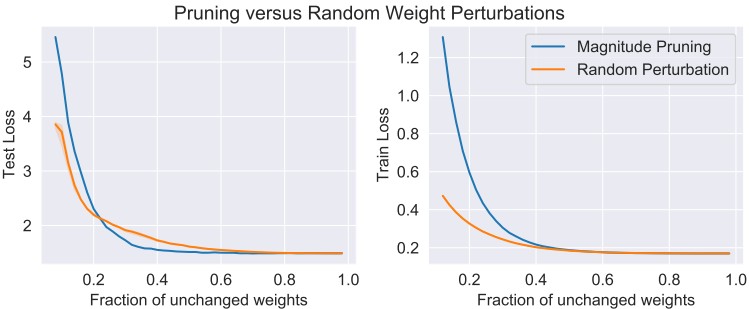

Figure 5: **Pruning affects models differently than random perturbations.** Here we compare pruning of weights and randomly perturbing the same weights by the same amount. We study a CNN on CIFAR100. Generally speaking, pruning will have a larger negative impact on a model's loss than randomly perturbing the same weights by the same amount. Similarly, as in the original experiments, we find that pruning has a smaller negative impact on the test loss than randomly perturbing as long as a moderate number of parameters are being perturbed. For larger fractions of perturbed parameters, pruning has a larger negative impact on the test loss than random perturbations.

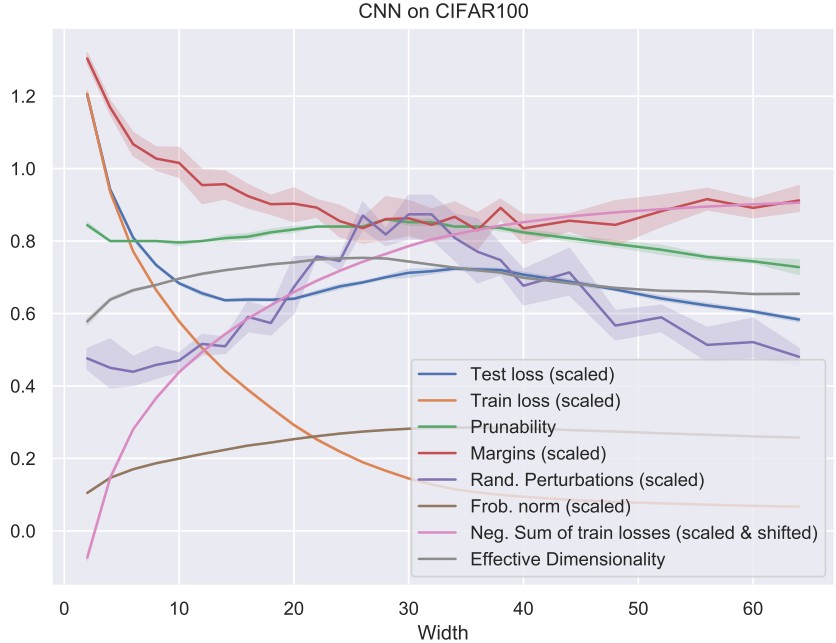

(a) Comparison of prunability and baseline measures in a double descent setting. As indicated, some measures were scaled to fit onto the same y-axis. This Figure is the extended version of Figure 1.

| Kendall's $\tau$ | Test Loss | Test Error |
|---|---|---|
| Prunability | **0.3613** | 0.1277 |
| Eff. Dim. | -0.0197 | -0.0753 |
| Random Perturbation | 0.1398 | -0.2166 |
| Frobenius Norm | 0.0996 | -0.5367 |
| Margins | 0.1296 | **0.4258** |
| -Sum of Training Losses | -0.2890 | -0.9381 |

(b) Kendall's rank correlation ($s_\tau = 0.06$) between generalization measures and test loss/test error. Higher values are better.

Figure 3: **Prunability is correlated with test loss in double descent setting:** Across a set of convolutional networks of varying width trained on CIFAR-100, we show that prunability captures double descent behavior better than a previously proposed metric *Effective Dimensionality* – which is based on the eigenspectrum of the Hessian of the training loss of the model – and other strong baselines.

## C    PAC-BAYES BOUND

We use the following, simplified formulation of McAllester (1999)'s dropout bounds which is also used in Jiang et al. (2020):

**Theorem 1** *For any $\delta > 0$, distribution D, prior P, with probability $1 - \delta$ over the training set, for any posterior Q the following bound holds:*

$$\mathbb{E}_{\boldsymbol{v} \sim Q}[L(f_{\boldsymbol{v}})] \leq \mathbb{E}_{\boldsymbol{v} \sim Q}[\hat{L}(f_{\boldsymbol{v}})] + \sqrt{\frac{KL(Q||P) + \log(\frac{m}{\delta})}{2(m-1)}}$$

For the reader's convenience we briefly reproduce the derivation of the dropout bound in McAllester (2013).

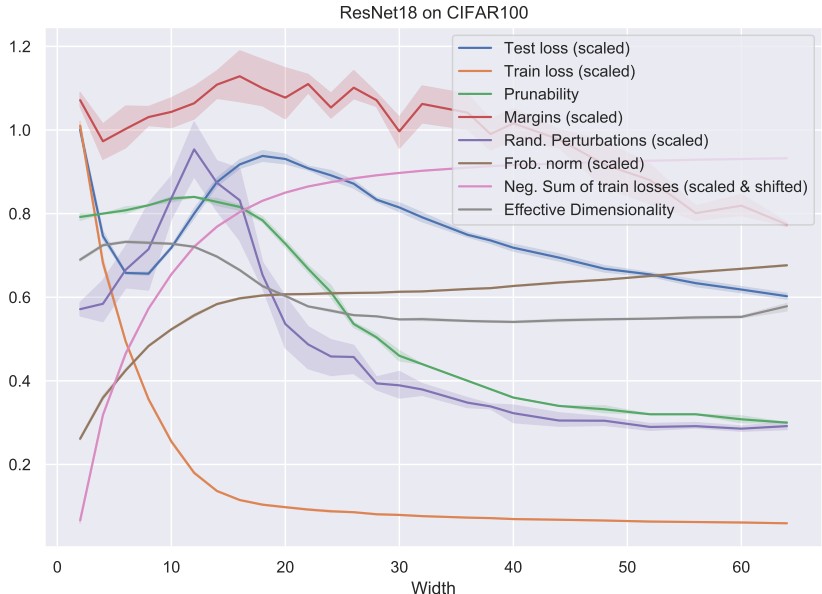

(a) Comparison of prunability and baseline measures in a double descent setting. As indicated, some measures were scaled to fit onto the same y-axis.

| Kendall's $\tau$ | Test Loss | Test Error |
|---|---|---|
| Prunability | 0.4812 | **0.7556** |
| Eff. Dim. | 0.1272 | 0.5077 |
| Random Perturbation | 0.4532 | 0.7370 |
| Frobenius Norm | -0.4487 | -0.9192 |
| Margins | **0.5589** | 0.3883 |
| -Sum of Train Losses | -0.4728 | -0.9393 |

(b) Kendall's rank correlation ($s_\tau = 0.06$) between generalization measures and test loss/test error. Higher values are better.

Figure 4: **Prunability is correlated with test loss in double descent setting:** Across a set of ResNet18s of varying width trained on CIFAR-100, we show that prunability captures double descent behavior better than a previously proposed metric *Effective Dimensionality* – which is based on the eigenspectrum of the Hessian of the training loss of the model – and is competitive with other strong baselines.

To draw a sample $\omega$ from $Q$ means first drawing a sparsity pattern $S \sim S_\alpha$, where $\alpha$ is the dropout probability, and a noise vector $\epsilon \sim \mathcal{N}(0,1)^d$. Then, we construct $\omega$ as $s \circ (\boldsymbol{w} + \epsilon)$, $\circ$ being the Hadamard product: $(s \circ w)_i := s_i w_i$. Similarly, a sample from $P$ is $s \circ \epsilon$. We denote by $S_\alpha$ the distribution on the $d$-cube $\mathcal{B}$ of sparsity patterns. We then have $\mathbb{E}_{\boldsymbol{v} \sim Q}[f_{\boldsymbol{v}}] = \mathbb{E}_{s \sim S_\alpha, \epsilon \sim \mathcal{N}(0,1)^d}[f_{s \circ (\boldsymbol{w}+\epsilon)}]$. We derive $KL(Q||P)$ as follows:

$$KL(Q||P) = \mathbb{E}_{s \sim S_\alpha, \epsilon \sim \mathcal{N}(0,1)^d}\big[\ln \frac{S_\alpha(s)e^{-\frac{1}{2}||s \circ \epsilon||^2}}{S_\alpha(s)e^{-\frac{1}{2}||s \circ (\boldsymbol{w}+\epsilon)||^2}}\big]$$

$$= \mathbb{E}_{s \sim S_\alpha}\big[\frac{1}{2}||s \circ \boldsymbol{w}||^2\big]$$

$$= \frac{1-\alpha}{2}||\boldsymbol{w}||^2$$

We can directly obtain a bound for prunability from this, by searching for the largest $\alpha$, s.t. $\mathbb{E}_{\boldsymbol{v} \sim Q}[\hat{L}(f_{\boldsymbol{v}})] < \hat{L}(f_{\boldsymbol{w}}) \times (1 + \beta)$, where $\boldsymbol{w}$ are the weights learned during training of the network. We have to account for the fact that we search for $\alpha$ in our bound. We make use of the fact that we're searching over a fixed number $c$ of $\alpha$s and use union bound in the bound which will change the log term in the bound to $\log(\frac{cm}{\delta})$. Gives us the following bound:

$$\mathbb{E}_{\boldsymbol{v} \sim Q}[L(f_{\boldsymbol{v}})] \leq \mathbb{E}_{\boldsymbol{v} \sim Q}[\hat{L}(f_{\boldsymbol{v}})] + \sqrt{\frac{\frac{1-\alpha}{2}||\boldsymbol{w}||_2^2 + \log(\frac{m}{\delta}) + 5}{2(m-1)}}$$

