# OpenReview forum: "Robustness to Pruning Predicts Generalization in Deep Neural Networks"
_ICLR.cc/2021/Conference — Reject_

### Official Review · AnonReviewer3 · 2020-10-26
**I think that the authors propose a sensible research direction, but the presented results might be insufficient for justifying the introduction of an nth complexity measure in the generalization debate.**

**Rating:** 5
**Confidence:** 3

**Review:**

In the present work, the authors tackle the highly debated (and sometimes confusing) problem of finding a good simplicity/complexity measure able to predict generalization performance of deep networks. A novel measure called 'prunability' is introduced and compared with some of the many alternatives in the literature. This property measures how networks are able to retain low training loss when a fraction of the weights is set to zero, and is clearly related to common training practices (e.g. dropout) that seems to yield better generalization performance in practice. The experimental settings and the evaluation methods for this new metric are inspired by recent extensive studies on deep networks performance. The authors are able to show that prunability is in fact associated with good generalization and seems able to capture some non-trivial phenomena (double-descent), but they also find it to be inferior to pre-existing (margin based) measures. Moreover, the close relationship to perturbation robustness and flatness measures is investigated, but the results are not fully conclusive.
In my opinion, the idea behind this complexity measure makes a lot of sense, but: 1) it is already used explicitly in dropout, so I don't see how it could inspire the development of better training heuristics. 2) It is not easier to measure that most other complexity measures (in the random version). 3) It is closely related to pre-existing metrics. 4) More importantly, it is not more predictive of generalization than some alternatives already discussed in the literature. Therefore, I think that this works lacks at least one strong point that could motivate the inclusion of this latest complexity measure into the generalization debate.

---

> ### Author Response · Authors · 2020-11-17
> **Response to AnonReviewer3**
>
> Dear reviewer,
>
> Thank you for taking the time to engage with our paper and the encouraging comments. We are happy to hear that you think we’re pursuing a promising research direction.
>
> To summarize our response: we believe that our proposed generalization measure is worth including in the generalization measure debate precisely because it is competitive with some of the few generalization measures that actually *work well* and because it is sufficiently different from existing measures, thus adding another piece to the puzzle of model generalisation.
>
> In the following, we would like to address the questions you raise. We would greatly appreciate to hear your thoughts on our replies!
>
> Particularly, you write:
> * “In my opinion, the idea behind this complexity measure makes a lot of sense, but: 1) it is already used explicitly in dropout, so I don't see how it could inspire the development of better training heuristics. 2) It is not easier to measure that most other complexity measures (in the random version). 3) It is closely related to pre-existing metrics. 4) More importantly, it is not more predictive of generalization than some alternatives already discussed in the literature. Therefore, I think that this work lacks at least one strong point that could motivate the inclusion of this latest complexity measure into the generalization debate.”
>
> We are thinking about these points as follows:
> 1) It’s true that random pruning  – in the form of dropout –  is already being used during training. However, randomly dropping out a fixed percentage of a model’s weights during training only indirectly optimizes prunability, leaving open the possibility of more effective approaches. For example, two training heuristics which more directly optimize prunability would be to drop out the highest-magnitude weights, or to increase the percentage of dropped out weights over the course of training. While this is not the main focus of our work, we do see our contribution as a stepping stone towards further research on such methods.
> 2) We agree with your point that prunability is similarly expensive to measure as some of the other generalization measures we evaluate. It is however, much less expensive than the best-performing, margins-based generalization measure which requires the training of a large number of models to train the linear regression on which this measure is based.
> 3) AnonReviewer2, nicely summarizes our view on this. We would argue that our measure is novel in a number of ways: first, we establish a particular connection between pruning and generalization that has not been described before; second, we establish a way of formulating a generalization measure based on a model’s number of parameters – which has proved difficult so far; third, we do show some preliminary evidence that our measure does seem to measure something different -- and more useful -- than the seemingly related random perturbation robustness.
> 4) Lastly, we agree that prunability is not the single best known generalization measure. Finding the most predictive generalization measure is one of the goals of the generalization literature broadly. Another important goal (one which is instrumental to the first, but also interesting as a scientific question in its own right) is to _understand_ generalization. It is possible to contribute to the latter without aiming for the first of these goals. We see our study of prunability as just such a contribution.  Prunability is a generalization measure that is competitive with some of the strongest known generalization measures – and seems different from them. It clearly outperforms random perturbation robustness, for instance, which in turn was one of the strongest out of the 40 or so studied in the Fantastic Generalization measures paper. Note that while the margin-based generalization measure is the most predictive baseline measure we study, it also has its disadvantages. It is, for instance, data set specific and would thus have to be retrained to be used on a new data set. See also AnonReviewer2’s framing of topic, with which we very much agree.
>
> To summarize, we believe that prunability is in fact a very strong (although not the strongest), novel generalization measure and that it would be valuable to add to the discussion around generalization measures.
>
> Thanks again,
> Authors of Paper3241

---

> > ### Comment · AnonReviewer3 · 2020-11-24
> > **Response to the authors**
> >
> > I would like to thank the authors for trying to provide concrete answers to all the concerns raised in the review process. Overall, I completely agree with authors and reviewers reiterating the fact that this is an empirical study, and I value the type of contribution these works can give in understanding such complex phenomena.
> >
> > As this point frequently came out in the above discussion, I think it would be nice to somewhat clarify the connection between the employment of dropout during training and prunability at the end of training (random or magnitude based). Since the authors seem to believe that there is only an indirect connection, maybe studying this relationship in a simple experimental setup could highlight the novelty of the introduced measure or at least reconnect it with the known heuristics.
> >
> > I think that the already added experiments in the revised version certainly strengthen the numerical results and clarify the value of this new measure in comparison to the previous ones. Moreover, I think the authors make good points in the response to my review (2-3-4) and that even if this work cannot put an end to the generalization debate, it still introduces a new thought provoking ingredient. I will therefore raise my score by one point to a 6.

---

> > > ### Author Response · Authors · 2020-11-24
> > > **Response to AnonReviewer3**
> > >
> > > Dear reviewer,
> > >
> > > We are happy to hear that you find our additional experiments convincing, that we were able to address your concerns and that you will raise your score to a 6. Thank you very much.
> > >
> > > To communicate our current understanding of the relation between prunability and dropout more clearly, we've added an additional comment at the end of the Related Work section in our latest revision. We agree that studying this relation more closely would be an interesting direction for future research.
> > >
> > > Thank you,
> > >
> > > Authors of Paper3241

---

### Official Review · AnonReviewer2 · 2020-10-28
**Solid empirical observations connecting prunability to generalization in deep learning**

**Rating:** 7
**Confidence:** 4

**Review:**

## Paper summary

In order to understand why deep networks generalize well, this paper proposes "prunability" as an empirical measure that can be predictive of the generalization. Prunability is roughly the smallest _fraction_ (i.e., $\in [0,1]$) of parameters that can be retained, while zeroing out everything else, without increasing the model's training loss by too much. The authors experimentally demonstrate the predictive ability of this measure in three ways.

1. They consider **the large-scale empirical framework of Jiang et al' 2019** where one computes different statistics about "generalization vs. the measure" from a large pool of trained models (viz., DEMOGEN dataset from Jiang et al' 2018). They compare prunability against four other existing measures:  Frobenius norms, random perturbation robustness (a flatness-based measure), Normalized margins (a layer-wise margin measure), the average loss across training (a speed-of-optimization based measure).

 Then they compute three different previously-proposed statistics:
   - a) Kendall's rank correlation coefficient (Jiang et al' 2019) which tells us how well the measure can rank the models: **here prunability performs better than the flatness measure, and much better than the norm and speed measures.** (However, normalized margins outperform everything)
   - b) adjusted $R^2$: here prunability performs just as well as other measures (although normalized margins outperform again.)
   - c) a conditional mutual information (CMI) term Jiang et al 2019 that tells us whether the measure has a causal role in the generalization behavior: prunability's CMI is pretty low revealing poor causal connections.

2. They conduct Maddox et al., 2020's experiment where one evaluates the measure for varying widths to see whether it can **capture the double descent behavior of the test loss**. They observe that prunability does show a double descent behavior, while Maddox et al., 2020's flatness-based "effective dimensionality" only shows an "ascent-descent" behavior.

3. Finally they demonstrate that **prunability captures something different from all the other measures.** They do this by showing that there is little causal connection between prunability and all other measures except the flatness-based random-perturbation-robustness measure. Then they go on to show that pruning and random perturbations affect models differently. Notably, pruning can lower the test loss of the network while random perturbations always only increases the test loss.


## Strengths

1. The idea that compressibility relates to generalization is not new and has been theoretically quantified via generalization bounds. However, such bounds are still parameter count dependent and/or they're computed for a handful of models and it's not clear how well they correlate with generalization.  The paper takes an orthogonal direction towards relating compressibility and generalization: it pins down an empirical measure of compressibility, and then
provides three sufficiently different kinds of arguments to demonstrate the usefulness of that metric. Although the experiments used within these arguments in themselves are not novel, I think the fact the metric holds up in all these tests is interesting.

2. The way this paper quantifies prunability --- in terms of the fraction of parameters --- is simple and also somewhat thought-provoking (why should generalization be related to the fraction of parameters?).


3. The paper is honest and rigorous in terms of the values it reports: prunability is not the best of all metrics, and the paper is transparent about it. The paper also gives sufficient credit to work that it builds upon. The writing is smooth.

## Weaknesses

4. Given that this is a purely empirical paper, I'd have appreciated if the observations made in the double-descent experiment and "the effect of pruning-vs-perturbation on test loss" experiment, were also shown in at least one other dataset/architecture.


## Overall opinion

This paper provides multiple different pieces of evidence backing its claim that prunability predicts generalization. These empirical observations are rigorous and would be valuable in understanding the generalization puzzle. This way of quantifying prunability might also open up new theoretical questions. Hence, I think this is a good paper worth publishing.

## Clarification questions

5. Could you explain why the last few experiments are reported in terms of the cross-entropy loss? Is it because the corresponding plots for the 01 error do not show as much different between the lines? Nevertheless, I feel that it'd be nice to have those plots in the paper too (no pressure to produce those plots during rebuttal).

6. Could you clarify the claims in the first paragraph of 5.4.1? Specifically
>"prunability is highly informative of generalization across all of our evaluation metrics  outperforming random perturbation robustness, the training loss itself and the Frobenius norm measures"

seems to contradict later observations that on some evaluation metrics they are all just as good as each other on adjusted $R^2$.

### Minor suggestions

- Under Table 1 and 2 would be nice to remind the reader as to whether larger values indicate better predictivity or not.
- Is the usage of adjusted $R^2$ inspired by Jiang et al., 2018? If so, would help to cite them appropriately.
- Page 12 typo "Vargins" --> "Margins"

### Suggested citation

"On the importance of single directions for generalization" Ari S. Morcos, David G.T. Barrett, Neil C. Rabinowitz, Matthew Botvinick, ICLR 2018 https://arxiv.org/abs/1803.06959   -- they empirically study how generalization is related to how many hidden units you can zero out. (Certainly not the same as what the submission suggests, but I think worth citing.)

### References
- Yiding Jiang, Behnam Neyshabur, Hossein Mobahi, Dilip Krishnan, and Samy Bengio. Fantastic
generalization measures and where to find them. arXiv preprint arXiv:1912.02178, 2019.

- Yiding Jiang, Dilip Krishnan, Hossein Mobahi, and Samy Bengio. Predicting the generalization gap
in deep networks with margin distributions. arXiv preprint arXiv:1810.00113, 2018.

- Wesley J. Maddox, Gregory Benton, and Andrew Gordon Wilson. Rethinking parameter counting
in deep models: Effective dimensionality revisited, 2020.


**Update:** The authors clarified all my questions very well. They also added an extra plot for the double descent experiment on a different architecture (ResNet). Although I feel a bit lukewarm about the added plot (in that the double descent phenomenon is only somewhat weakly reflected by their empirical measure), I've increased my confidence score from 3 to 4 to appreciate their efforts in addressing my concerns. Good luck to the authors!

---

> ### Author Response · Authors · 2020-11-17
> **Response to AnonReviewer2**
>
> Dear reviewer,
>
> Thank you for your review – we are happy to hear that you enjoyed our paper. We very much agree with your assessment of the strengths and weaknesses of our paper.
>
> We’d like to briefly address the questions you raise. We would be very interested to hear what you think about our responses.
>
> You write:
> * “Given that this is a purely empirical paper, I'd have appreciated if the observations made in the double-descent experiment and "the effect of pruning-vs-perturbation on test loss" experiment, were also shown in at least one other dataset/architecture.”
>
> Yes, we agree that it makes sense to extend the double descent and pruning-vs-random-perturbation experiments. We have added one additional experiment for each of these in Appendix B of the paper and we find that the results of the additional experiments are consistent with the results of the previous experiments. Thank you for the suggestion.
>
> You ask:
> * Could you explain why the last few experiments are reported in terms of the cross-entropy loss? Is it because the corresponding plots for the 01 error do not show as much different between the lines? Nevertheless, I feel that it'd be nice to have those plots in the paper too (no pressure to produce those plots during rebuttal).
>
> The experimental set up of our double descent experiment is directly taken from Maddox et al., 2020. In this setting, we study the cross-entropy rather than the 1-0 loss because that is the metric for which the double descent has been observed. While this is not the main focus in these experiments, we also report the correlation between the generalization measures and the test error in Figure 1b).
>
> You ask:
> * Could you clarify the claims in the first paragraph of 5.4.1? Specifically "prunability is highly informative of generalization across all of our evaluation metrics outperforming random perturbation robustness, the training loss itself and the Frobenius norm measures" seems to contradict later observations that on some evaluation metrics they are all just as good as each other on adjusted R2
>
> Thanks for pointing out the imprecision of our statement with regards to the performance in terms of Adjusted R2 of the different measures. Indeed, the norm-based measure seems to outperform the other mentioned measures in terms of Adjusted R2. We’ve added a clarifying comment to the paper, thanks.
>
> Does this make things clearer to you?
>
> Thank you, we will add the citation you suggest and do the minor fixes in the next revision of the paper.
>
> Thanks again,
>
> Authors of Paper3241
>
> Wesley J. Maddox, Gregory Benton, and Andrew Gordon Wilson.  Rethinking parameter counting in deep models: Effective dimensionality revisited, 2020.

---

> > ### Comment · AnonReviewer2 · 2020-11-17
> > **Thanks for clarifying my concerns**
> >
> > Thank you for spending the time and efforts to add the experiments you asked for. Based on these updates and the discussion so far, I'm going to increase my confidence score to a 4. I will increase my score up to one more point based on the discussions with the other reviewers.
> >
> > As for the double descent curve, I agree that there are some qualitative differences between prunability and effective dimensionality even for ResNet18. Although, these effects are a bit hard to appreciate in the added plot (I don't see the first descent, but I do see an initial "convex" curvature that prunability has but effective dimensionality doesn't). Nevertheless, I appreciate this added plot. I'll also wait for the baselines that Anonreviewer4 has asked as I think that'd be an illuminating addition.
> >
> > For future versions of this paper, I feel that this section is something that you could consider fleshing out further (try other datasets, try other ways of increasing parameter count etc.,) -- however this does not affect my scoring anymore, so please feel free to ignore this suggestion for now.
> >
> >
> > Thanks for your clarification regarding the cross-entropy loss. That makes sense and sounds ok to me.
> >
> > As for the last point, I'm not sure I understand the clarification. I still see the line "We find
> > that prunability is highly informative of generalization across all of our evaluation metrics, outperforming random perturbation robustness, the training loss itself and the Frobenius norm measures" which contradicts the table. Could you clarify further?

---

> > > ### Author Response · Authors · 2020-11-23
> > > **Thank you for your comment**
> > >
> > > Dear reviewer,
> > >
> > > Thank you very much for your feedback and for increasing your confidence score!
> > >
> > > It seems indeed like we accidentally didn’t include the correction for the phrase you quote. We have now changed it to: “In particular, it outperforms random perturbation robustness, the training loss itself and the Frobenius norm measures in terms of Kendall's rank correlation coefficient.”
> > >
> > > We added the additional experiments AnonReviewer4 suggested to the latest revision of the paper (see Figure 1 and Appendix B). We would be very interested in hearing your thoughts on them.
> > >
> > > Thank you,
> > >
> > > Authors of Paper 3241

---

### Official Review · AnonReviewer1 · 2020-10-28
**Interesting ideas with potential but weak theoretical justifications**

**Rating:** 5
**Confidence:** 4

**Review:**

The authors consider the problem of estimating generalization in deep neural networks, and propose a measure based on the ability to set a fraction of the neural network weights to zero (prunability). The authors introduce some theoretical motivation based on the PAC-Bayesian framework, and perform an empirical evaluation based on a set of convolutional networks trained on the CIFAR-10 dataset.

The problem of understanding generalization in deep neural networks is a fundamental problem of deep learning, and the results obtained by the authors present a potentially interesting perspective on the study of generalization in deep neural networks. The paper is well-written, and presents some interesting empirical results. However, the theoretical contribution is not  particularly novel, and is not complete enough to justify all the measures evaluated in paper. Due to this issue, I feel that the paper only has limited impact. I detail my comments on the theoretical and empirical results of the paper below.

On the theoretical side, the contribution for the case of “random pruning” is minor and is a straightforward extension of known results. Indeed, “random pruning” is similar to many other well-studied random perturbation schemes, and corresponds exactly to the case of dropout, a well-understood method. On the other hand, there does not appear to be any bound provided for the case of “magnitude pruning”, and it is not immediately obvious why such a measure should lead to a generalization bound. Indeed, existing work on using pruning and compression for measuring generalization (Arora et al. 2018, Zhou et al. 2019) establish bounds on the *pruned* network, and not the original network. Establishing a bound in terms of the “magnitude pruning” measure would be an interesting contribution, as handling such non-random modifications has been a challenge in the community. However, the authors do not seem to make any attempt at providing such a bound or a heuristic argument for such a bound to hold. Finally, the choice of measuring the prunability of a network as a proportion requires more careful justification in the context of magnitude pruning. Indeed, for “random pruning”, the proportion is easily interpreted as a magnitude of noise injected. However, in the “magnitude pruning” setup, with the parallels the authors draw to Occam’s razor and compression ideas, it is the absolute number of parameters which is more theoretically relevant than the proportion of parameters which can be eliminated: having 1 million parameters where half can be eliminated is still more complex than only having 100,000 parameters (where none can be eliminated).

On the empirical side, I found that the methodology was clear and well-adapted for the “random pruning” method, which is closely related to drop-out and other random perturbation ideas. With the addition of networks of different depth in the set of networks used for evaluation, the empirical methodology also seems appropriate for the “magnitude pruning” method and the parallels the authors draw to compression. However, it is not obvious to me that the empirical results support that connection, as we know that in architectures which vary substantially in size (e.g. MobileNet or EfficientNet), the compressibility (i.e. the proportion of parameters which can be pruned) is directly related to the size of the network (see e.g. Gupta et al. 2017), and I feel that this is a further indication that the choice of using the proportion of pruned parameters should be more carefully discussed.

The presentation of the empirical result could be improved by including (either in the main text or the appendix) the standard errors for the measured correlation for all measures, and ensuring that the tables are formatted similarly to the ICLR template for better readability (the latex package booktabs can be used for this purpose).

Other notes: please ensure that you cite published versions of papers when they are available (instead of the arxiv pre-prints). For example, Arora et al. 2018 appeared at ICML 2018, and Zhou et al. 2018 [sic] appeared at ICLR 2019 (there are many other such cases in the references).

=====================

Edited after author response: I thank the authors for their considerate responses. Overall, my opinion remains mostly unchanged, and I share similar opinions to reviewer 3 and 4 that although the proposed idea is interesting and intriguing, the paper is not quite ready at this point. I would like to see the authors present either: 1) stronger empirical evidence for the importance of their metric or 2) a more solid theoretical foundation of the measure they propose.

---

> ### Comment · AnonReviewer2 · 2020-11-17
> **Some comments on the "theoretical contribution"**
>
> I cannot speak on behalf of the authors and I'd like to see their response to your review. But FWIW (as a fellow reviewer), I wanted to discuss your main criticism that "the theoretical contribution is not particularly novel, and is not complete enough to justify all the measures evaluated in paper".
>
> First, my understanding is that this is *a purely empirical paper*. The paper to me seems very clear about this --- I didn't see any theoretical contributions that they claimed is theirs.  When they mention the theorems, I felt that the intention was to just discuss *existing* theoretical notions connecting some notion of prunability to generalization. This is evident to me through the following sentences preceding the theorem: "The connection between dropout and generalization has been studied in (McAllester, 1999)", "First, it is well-established that we can remove large fractions...", "training models using dropout, i.e. random pruning, is widely known to improve generalization..." So I feel that this criticism that their theoretical contribution is not novel may be a bit of a strawman here since they didn't claim anything of that sort in the first place? Or maybe I'm missing --- if you felt that there are places where they implied so, could you point out? Hopefully, the authors would find that useful to set the readers' expectations right.
>
> Second, regarding the "theoretical justification" of their empirical measure, I think a paper is valuable even if it can come up with an empirical measure that seems to predict generalization in practice, and if the paper conducts rigorous experiments to validate its predictive power. Future work can then study the (interesting) question of "why does this (strange) measure predict generalization?" by connecting this measure to generalization via a generalization bound. One such example of a purely empirical paper I could think of was the "On the importance of single directions for generalization" paper.  https://arxiv.org/abs/1803.06959
>
> Sure, there may be criticisms about whether their experiments in this paper really are convincing of the predictive power, and those criticisms may be valid. But I would be hesitant to agree with the criticism that this paper should theoretically justify their measure.
>
> But based on your criticism, I do think that the paper can (a) place those theorems under a  "background/preliminary" section to be even more clear to the readers and (b) mention somewhere that it's not theoretically clear why the fraction of parameters can predict generalization.

---

> > ### Author Response · Authors · 2020-11-23
> > **Response to AnonReviewer2's comment on AnonReviewer1's review**
> >
> > Dear reviewer,
> >
> > Thank you very much for your comment.
> >
> > We very much agree with the points you raise. To make our position with regards to our theoretical contribution even more clear, we extended the conclusion to state “Given the strong empirical performance of prunability – and given that the theory behind its success is not yet well understood – this measure of model complexity may be of use to the construction of future generalization bounds”.
> >
> > Thank you for the suggestion,
> >
> > Authors of paper 3241

---

> ### Author Response · Authors · 2020-11-17
> **Response to AnonReviewer1 – Part I**
>
> Dear reviewer,
>
> Thank you for your review and your thoughtful comments. We’d like to address some of the points you raise. Please let us know whether our replies make sense to you or whether you would like further clarifications.
>
> With regards to our theoretical justification/motivation you write:
> * “On the theoretical side, the contribution for the case of “random pruning” is minor and is a straightforward extension of known results. [...] Establishing a bound in terms of the “magnitude pruning” measure would be an interesting contribution, as handling such non-random modifications has been a challenge in the community. However, the authors do not seem to make any attempt at providing such a bound or a heuristic argument for such a bound to hold.”
>
> Our view on this is the following:
> * We agree that our theoretical justification for robustness to pruning is not our main contribution. We do, however, strongly believe that it is highly valuable to empirically identify new aspects of deep learning models that relate to generalization. We think that the explanation AnonReviewer2 gives under 1) in the Strengths section nicely summarizes our position on this.
>
> Furthermore, you write:
> *  “Finally, the choice of measuring the prunability of a network as a proportion requires more careful justification in the context of magnitude pruning. Indeed, for “random pruning”, the proportion is easily interpreted as a magnitude of noise injected.”
>
> Our view on this is the following:
> * We agree with you that we could have more clearly motivated the step from studying random pruning to magnitude pruning. Magnitude pruning can remove a greater number of model weights without affecting performance, and can therefore be considered a more effective pruning strategy. We hypothesized that using a more effective pruning method might yield a measure of prunability that better predicts generalization. We will add this to a revision of the paper, thank you for the suggestion.
>
> With regards to our empirical contribution you write:
> * “On the empirical side, I found that the methodology was clear and well-adapted for the “random pruning” method, which is closely related to drop-out and other random perturbation ideas. With the addition of networks of different depth in the set of networks used for evaluation, the empirical methodology also seems appropriate for the “magnitude pruning” method and the parallels the authors draw to compression. However, it is not obvious to me that the empirical results support that connection, as we know that in architectures which vary substantially in size (e.g. MobileNet or EfficientNet), the compressibility (i.e. the proportion of parameters which can be pruned) is directly related to the size of the network (see e.g. Gupta et al. 2017), and I feel that this is a further indication that the choice of using the proportion of pruned parameters should be more carefully discussed.”
>
> Our view on this is the following:
> * We agree that the compressibility of models of different sizes varies a lot. While we do not compare different architectures of different sizes, we do compare models of the same architecture with widely different numbers of parameters by varying both the depth and the width. Across these models of different sizes prunability does indeed seem predictive of generalization.
>
> [we continue our response below]

---

> > ### Author Response · Authors · 2020-11-17
> > **Response to AnonReviewer1 – Part II**
> >
> > [see the previous comment for the first part of our response]
> >
> > You write:
> > * “However, in the “magnitude pruning” setup, with the parallels the authors draw to Occam’s razor and compression ideas, it is the absolute number of parameters which is more theoretically relevant than the proportion of parameters which can be eliminated: having 1 million parameters where half can be eliminated is still more complex than only having 100,000 parameters (where none can be eliminated)”
> >
> > Generally, it seems difficult to reason about model complexity in terms of the numbers of their parameters, given that parameter counting has been shown to be not very informative of model’s generalization power (see Jiang et al., 2020, for instance). Furthermore, we do study model’s of different sizes (we compare models of different depth and width) and across these the fraction of parameters that can be kept is indeed predictive whereas the absolute number of parameters is not. Lastly, even if we assume that looking at the number of parameters is informative, it is not self-evident that a larger but highly sparse model would be more complex than a smaller, dense model according to many learning-theoretic measures. For example, in the random-pruning setting, it is not obvious to us that the set of functions which can be computed by a given architecture with $n$ parameters is necessarily smaller than the set of functions that can be computed by an architecture with $2n$ parameters, of which any $n$ may be removed without affecting the function’s loss.
> >
> > We have updated our citations to refer to the published versions of resources where applicable. Additionally, for the reader’s convenience, we’ve added URLs for all resources.
> >
> > Thank you for the suggestions of including the standard errors, we’ll adapt the next revision of the paper accordingly.
> >
> > Thanks again,
> >
> > Authors of Paper3241
> >
> > Yiding Jiang*, Behnam Neyshabur*, Hossein Mobahi, Dilip Krishnan, and Samy Bengio.  Fantastic generalization measures and where to find them. In International Conference on Learning Representations, 2020

---

### Official Review · AnonReviewer4 · 2020-10-28
**Pruning as Generalisation Measure**

**Rating:** 5
**Confidence:** 4

**Review:**

The paper proposes a novel generalisation measure, i.e., measurement that indicates how well the network generalises, based on pruning. The idea is to measure the fraction of the weights that can be pruned (either randomly, or based on the norms) without hurting the training loss of the model. The paper provides thorough discussion of the related methods and motivates the measure in multiple ways. Further, the authors show empirical evidence for the correlation of the pruning robustness to the generalisation ability of networks, based on the paper by Jiang et al., 2019 and dataset (updated with additional models) provided in the paper.

The paper is well written and states a clear goal of introducing and proving a generalisation measure based on pruning. The authors provide a nice discussion and lots of empirical evidence. Nonetheless, several points of motivation for the measure seem unclear to me:

- the authors refer to Jiang et al., 2019 saying that there are lots of generalisation measures that correlate with performance, but that they fail to explain the test performance - which calls for another measure. If the experiments shown in Jiang et al. are demonstrating failure of explaining test performance, then the presented paper also does not provide many more evidences that the proposed measure is not failing.
- the motivation for pruning as a way to improve generalisation is connected to the training procedures - dropout and lottery ticket hypothesis. Nevertheless, none of these methods improve generalisation when applied on top of already trained network. I would not say that lottery ticket retraining can be classified as integration of pruning into optimisation, as well as dropout improves training only when all the network is used afterwards.

There are several failures, that make me believe that more work can improve the paper:
- The goal of the paper is to show a measure that will perfectly predict generalisation - but according to the experiments it can be outperformed by other measures on the presented dataset.
- The theoretical justification seem unclear to me: what is the goal of introducing the generalisation bound (moreover even Appendix does not have details of derivation of the presented formula) if the authors notify themselves right away that it is vacuous. The justification is to give an intuition of how pruning connects to generalisation. It is unclear to me though, how it can be concluded based on a vacuous bound?
- The idea to check the measures behaviour in double descent setup is very interesting, but only one measure is checked there and in a different experimental setup, without proper motivation for such change.
- Section5.4.3 attempts to analyse casual connection between existing measures, that seems to me unclear by motivation as well - one wants to see causal connection to generalization, not other measures. Especially, that the table3 is discussed only for the random perturbations measurement - still not providing an answer what type of the connection is there between two.

I would suggest to reject the paper, since the idea feels not being worked through enough.

Minor comments:

1 - it would be nice to have a discussion on the type of pruning used - does it somehow change the measurements in a predictable way?

2 - typo in the first sentence of section3 (twice “denote”)

3 - typo in the first sentence of the second paragraph of section3 (letter denoting data distribution)

4 - table1 misses highlights of the “winning” approaches

***

I would like to thank authors for accurate answers and a lot of work put on reworking the paper. Unfortunately, I still find my concerns about motivation for the metric valid, which together with the rather weak performance creates a problem for this paper. I highly encourage authors to continue the work and try to explain the reasons for this correlation and find justifications for usage of the metric.

---

> ### Author Response · Authors · 2020-11-17
> **Response to AnonReviewer4 – Part I**
>
> Dear reviewer,
>
> Thank you for taking the time to write a thorough review and the encouraging feedback. We are glad that you enjoyed our empirical contributions and we’d like to address the issues you mention. Please do let us know what you think about our replies below.
>
> With regards to the goals and results of our empirical study you write:
> * “If the experiments shown in Jiang et al. are demonstrating failure of explaining test performance, then the presented paper also does not provide many more evidences that the proposed measure is not failing.”
> * “The goal of the paper is to show a measure that will perfectly predict generalisation - but according to the experiments it can be outperformed by other measures on the presented dataset.”
>
> We don’t make the claim that our generalization measure is the single best generalization measure, nor is it our goal to find a generalization measure that perfectly explains generalization.
>
>  We do, however, strongly believe that it is highly valuable to identify new aspects of deep learning models that seem related to generalization, even if they don’t explain generalization entirely by themselves. We see our study of prunability as just such a contribution. We think that the explanation AnonReviewer2 gives under 1) in the Strengths section summarizes this idea very nicely.
>
> With regards to the motivation we provide in the paper you write:
> * “the motivation for pruning as a way to improve generalisation is connected to the training procedures - dropout and lottery ticket hypothesis. Nevertheless, none of these methods improve generalisation when applied on top of already trained network. I would not say that lottery ticket retraining can be classified as integration of pruning into optimisation, as well as dropout improves training only when all the network is used afterwards.”
>
> It is true that we do not provide a way of improving the generalization of models once they are trained but this is also not the ambition of our paper, as described above, nor is it a goal of existing work on generalization of neural networks.  Further, it is true that random pruning  – in the form of dropout –  is already widely used in training schemes. However, randomly dropping out a fixed percentage of a model’s weights during training only indirectly optimizes prunability, leaving open the possibility of more effective approaches. For example, two training heuristics which more directly optimize prunability would be to drop out the highest-magnitude weights, or to increase the percentage of dropped out weights over the course of training.
>
> With regards to the theoretical justification for studying pruning you write:
> * “The theoretical justification seem unclear to me: what is the goal of introducing the generalisation bound (moreover even Appendix does not have details of derivation of the presented formula) if the authors notify themselves right away that it is vacuous. The justification is to give an intuition of how pruning connects to generalisation. It is unclear to me though, how it can be concluded based on a vacuous bound?”
>
> The motivation behind our inclusion of the bound is to demonstrate that prunability is consistent with existing generalization theory: networks which are more prunable obtain lower PAC-Bayesian generalization bounds. So while the bound is, like most generalization bounds, likely to be vacuous in neural networks, it predicts that prunability should be predictive of generalization in terms of how it ranks models with different generalization performance. This ranking, then, is what we test in our experiments. We do agree that it would have been useful to include this explanation in the paper, so we added it to our latest revision in the paragraph below equation 1).
>
> With regards to our double descent experiments you write:
> * “The idea to check the measures behaviour in double descent setup is very interesting, but only one measure is checked there and in a different experimental setup, without proper motivation for such change.”
>
> This is a good point, we will evaluate additional baselines in the double descent setting and add them to the paper in the coming week. Thank you for the suggestion.
> We use two different experimental set ups that were designed for different purposes. One of them was designed to obtain a range of realistic models with a wide range of generalization performances by varying a number of relevant hyperparameters. The other setup was particularly chosen to obtain a test loss double descent which is only observed for a few particular configurations. The motivation then is: 1.) Prunability is informative of generalization in a general case and 2.) prunability is predictive of generalization even in a particularly challenging setting.
>
> [we continue our response below]

---

> > ### Author Response · Authors · 2020-11-17
> > **Response to AnonReviewer4 – Part II**
> >
> > [see the previous comment for the first part of our response]
> >
> > With regards to our experiments studying the conditional mutual information you write:
> > * “Section 5.4.3 attempts to analyse casual connection between existing measures, that seems to me unclear by motivation as well - one wants to see causal connection to generalization, not other measures. Especially, that the table3 is discussed only for the random perturbations measurement - still not providing an answer what type of the connection is there between two.”
> >
> > Our motivation for studying the conditional mutual information between prunability and the baseline generalization measures is to what extent prunability can already be explained by the baselines.
> >
> > Indeed, we do not provide a conclusive answer about the connection between prunability and random perturbation robustness. Rather, we point out that even though they seem related,  pruning and random weight perturbations seem to impact models’ losses quite differently. Based on this, we suggest that prunability deserves to be studied as a novel generalization measure – particularly given that it seems to outperform random perturbation robustness.
> >
> > We are working on the minor changes you suggest and will integrate them into a later revision of our paper, thank you!
> >
> > Thank you,
> >
> > Authors of Paper3241

---

> > > ### Comment · AnonReviewer4 · 2020-11-19
> > > **Response**
> > >
> > > Dear authors,
> > >
> > > Thank you for the answers to the concerns.
> > >
> > > I absolutely agree that the additional sources of information for understanding the properties of the deep learning models are very valuable to progress in the area, but it feels to me that this research can require some more insights to be published (as well as Reviewer2 noted, that the idea of connecting compression to generalization is not novel in itself).
> > >
> > > Regarding motivation, my largest concern is rather that I cannot relate the pruning _after_ training with the effect that dropout has. I do not see straightforward reasons to believe that ratio of "not needed" parameters is connected to the effect of forcing the network to learn examples using only part of parameters. As well as with lottery ticket hypothesis, which rather reflects the idea that usually a task can be learned with a smaller network - but it is more complicated process. Nevertheless, it again is not really connected to the generalization abilities of one or another variants.
> > >
> > > I understand your reasoning for the ranking of generalization abilities of networks with the help of PAC-bound, but if all the bound values are much larger than actual generalization loss of a network, then in general they can be ordered in any way - without any connection to the underlying ranking. Thank you for adding more explanations to the paper, I believe it is very useful.
> > >
> > > Great work with experiments and I am eager to see if they provide some solid conclusions.

---

> > > > ### Author Response · Authors · 2020-11-23
> > > > **Response to AnonReviewer4**
> > > >
> > > > Dear reviewer,
> > > >
> > > > Thank you for your response!
> > > >
> > > > In addition to the large scale study and the double descent experiment included in the first version of our paper, we’ve now also evaluated prunability in an additional double descent setting and evaluated additional strong baselines in these settings (see Figure 1 and Appendix B). The additional experiments seem to be consistent with the results of our large scale experiments: prunability is highly predictive of generalization and is competitive with some of the strongest known generalization measures.
> > > >
> > > > We agree that additional theoretical work will be required to properly understand prunability and its relation to other phenomena observed in deep neural networks. We further agree with the reviewer that our empirical results are not a corollary of known properties of dropout and iterative magnitude pruning; we will clarify in future revisions that while prior work on these techniques suggests that pruning and generalization are connected, our work presents a distinct and novel contribution in evaluating prunability in its own right. We strongly believe that our empirical results in their current form are valuable as a stepping stone and lay necessary groundwork for further theoretical work.
> > > >
> > > > We do hope that in light of our additional results and clarification of our narrative you would consider raising your score as we feel our work presents a valuable addition to the field.
> > > >
> > > > Thank you,
> > > >
> > > > Authors of Paper3241

---

### Decision · Program_Chairs · 2021-01-07
**Final Decision**

**Decision:**

Reject

**Comment:**

Summary:
The authors propose to predict a neural network classifier's
generalization performance by measuring the proportion of parameters
that can be pruned to produce an equivalent network (in terms of
training error). Experimental and theoretical evaluation are provided.


Discussion:
The overall opinion in reviews was that the idea is
potentially interesting, but needs to be pursued further before
publication, and that the empirical evaluation in particular was
lacking. That was followed by a detailed discussion, in which authors
were able to address a number of concerns, and have provided helpful
additional experiments.

Recommendation:
This is a potentially interesting paper that is not quite there
yet. Although reviewers have raised scores in discussion, the case for
acceptance would still be hard to make. I recommend to reject.

It looks like a reasonable  amount of additional work will turn this
from what is now on the weak end of borderline into a potentially strong
submission, especially given the thoughtful and thorough feedback from
reviewers. The next top-tier
conference deadline is not far away, and I encourage
the authors to incorporate the feedback fully and resubmit soon.
That being said, I agree with reviewers that the theory provided is,
at present, not strong. Also, a point that still seems to require work
is the relation between prunability and the use of dropout.

Note to authors and chairs:
AnonReviewer3 explicitly stated in
discussion that they would raise their score from 5 to 6, but the
change was not recorded in the system. My recommendation assumes their score
is 6.